# IndEcho study: cohort study investigating birth size, childhood growth and young adult cardiovascular risk factors as predictors of midlife myocardial structure and function in South Asians

Senthil K Vasan,[1,2] Ambuj Roy,[3,4] Viji Thomson Samuel,[5] Belavendra Antonisamy,[5] Santosh K Bhargava,[6] Anoop George Alex,[5] Bhaskar Singh,[6] Clive Osmond,[1] Finney S Geethanjali,[5] Fredrik Karpe,[2] Harshpal Sachdev,[7] Kanhaiya Agrawal,[5] Lakshmy Ramakrishnan,[4] Nikhil Tandon,[4] Nihal Thomas,[5] Prasanna S Premkumar,[5] Prrathepa Asaithambi,[5] Sneha F X Princy,[5] Sikha Sinha,[7] Thomas Vizhalil Paul,[5] Dorairaj Prabhakaran,[3,8] Caroline H D Fall[1]

For numbered affiliations see end of article.

**Correspondence to**
Dr Caroline H D Fall;
chdf@mrc.soton.ac.uk

## ABSTRACT

**Introduction** South Asians have high rates of cardiovascular disease (CVD) and its risk factors (hypertension, diabetes, dyslipidaemia and central obesity). Left ventricular (LV) hypertrophy and dysfunction are features of these disorders and important predictors of CVD mortality. Lower birth and infant weight and greater childhood weight gain are associated with increased adult CVD mortality, but there are few data on their relationship to LV function. The IndEcho study will examine associations of birth size, growth during infancy, childhood and adolescence and CVD risk factors in young adulthood with midlife cardiac structure and function in South Asian Indians.

**Methods and analysis** We propose to study approximately 3000 men and women aged 43–50 years from two birth cohorts established in 1969–1973: the New Delhi Birth Cohort (n=1508) and Vellore Birth Cohort (n=2156). They had serial measurements of weight and height from birth to early adulthood. CVD risk markers (body composition, blood pressure, glucose tolerance and lipids) and lifestyle characteristics (tobacco and alcohol consumption, physical activity, socioeconomic status) were assessed at age ~30 years. Clinical measurements in IndEcho will include anthropometry, blood pressure, biochemistry (glucose, fasting insulin and lipids, urinary albumin/creatinine ratio) and body composition by dual energy X-ray absorptiometry and bioelectrical impedance. Outcomes are LV mass and indices of LV systolic and diastolic function assessed by two-dimensional and Doppler echocardiography, carotid intimal-media thickness and ECG indicators of ischaemia. Regression and conditional growth models, adjusted for potential confounders, will be used to study associations of childhood and young adult exposures with these cardiovascular outcomes.

## Strengths and limitations of this study

► The strengths include the longitudinal study design with early growth data, which allows examination of the relationship of size at birth, childhood growth and cardiovascular disease risk markers in young adult life with myocardial structure and function in midlife in South Asians.

► The two cohorts represent both rural and urban populations, north and south India and diverse socioeconomic strata.

► The study will use harmonised methods to measure anthropometry, biochemical risk factors and to characterise lifestyle factors.

► A limitation of the study is attrition of the cohorts due to deaths and migration (mainly in childhood).

► Because this is a field-based study, echocardiography will be used, rather than cardiac MRI, which is the current gold standard method for measuring myocardial structure.

**Ethics and dissemination** The study has been approved by the Health Ministry Steering Committee, Government of India and institutional ethics committees of participating centres in India and the University of Southampton, UK. Results will be disseminated through scientific meetings and peer-reviewed journals.

**Trial registration number** ISRCTN13432279; Pre-results.

## INTRODUCTION
### Background and rationale

The emerging epidemic of cardiovascular disease (CVD) in transitioning populations means that low/middle-income countries

(LMICs) contribute a larger proportion to the global burden of CVD (~8–9 million deaths per year) than high-income countries (~5 million).[1–3] Migrants from LMICs to high-income countries experience an excess of CVD compared with indigenous populations.[4–6] A parallel increase in the prevalence of hypertension, type 2 diabetes (T2DM) and obesity, which are known risk factors for CVD is also observed in LMICs.[7 8]

Altered myocardial structure and function have received less attention as causes of cardiac death than ischaemic heart disease (IHD). Left ventricular hypertrophy (LVH) increases the future risk of heart failure and death[9] through volume overload, pressure overload and myocyte loss.[10 11] LVH is usually asymptomatic for several years before the development of congestive heart failure. Obesity, hypertension, T2DM and IHD initiate LV remodelling and dysfunction and enhance progression to heart failure.[11 12] The prospective Coronary Artery Risk Development in (Young) Adults (CARDIA) study showed that higher blood pressure, body mass index (BMI), waist circumference, cholesterol, triglyceride and glucose concentrations at age 18–30 years are risk factors for LV diastolic dysfunction 5–10 years later.[13] However, these traditional risk factors explain less than half the variability in echocardiographic parameters in population studies.[14 15]

South Asians develop CVD and T2DM, often in combination, a decade earlier than white Caucasians.[16 17] Several factors contribute to this premature disease: (1) lifestyle changes (the adoption of less healthy diets and reductions in physical activity, consequences of rapid socioeconomic transition), (2) a characteristic pattern of risk factors (low high-density lipoprotein (HDL) cholesterol, elevated lipoprotein (a) and high insulin resistance)[18 19] and (3) a 'thin-fat' body composition (low muscle mass but centrally adipose),[19 20] the latter two being evident even at birth.[21 22]

South Asian newborns have a low mean birth weight (2.6–2.7 kg) compared with white Caucasian babies[21 23] and frequently show suboptimal growth and weight gain during infancy.[24] Low birth and infant weight are risk factors for CVD mortality in adult life and have also been linked to a high risk of hypertension and T2DM.[25 26] It has been suggested that these associations reflect 'developmental programming', permanent structural and functional deficits in key metabolic organs (eg, liver, pancreas and kidneys) and tissues (eg, muscle) resulting from impaired nutrition during fetal and early childhood development.[27] Studies in India have linked lower birth weight to higher blood pressure, serum lipids and abnormal glucose tolerance in children and adults.[28–30] The risks associated with lower birth weight appear to be increased on subsequent exposure to greater child or adult weight gain. Upward crossing of BMI centiles in childhood and adolescence is associated with increases in several CVD risk factors such as higher adult cholesterol, triglyceride and proinflammatory marker concentrations and with an increased risk of obesity, impaired glucose tolerance or T2DM (figure 1), hypertension and metabolic syndrome.[29 31]

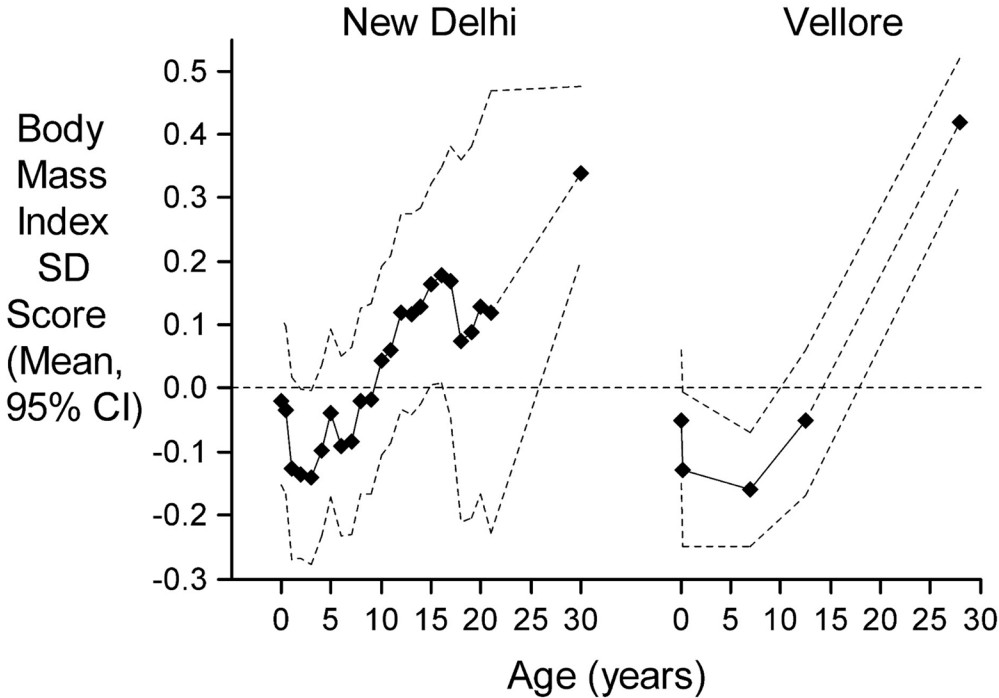

**Figure 1** BMI SD scores from birth to adulthood for participants in each cohort who developed IGT or diabetes (n=219/1562 in NDBC and 424/2218 in VBC) in adult life relative to the whole cohort (dashed zero line). BMI, body mass index; IGT, impaired glucose intolerance; NDBC, New Delhi Birth Cohort; VBC, Vellore Birth Cohort.

Early life factors may also influence cardiac structure and function. Growth restricted fetuses and newborns have impaired LV function,[32–34] thought to be caused by hypoxia, raised placental resistance and altered fetal circulation.[35] Studies relating birth weight to adolescent or adult LV size, in a variety of populations, have shown a positive association[36] or a non-significant inverse association.[37 38] In the latter two studies, low weight in infancy was associated with higher adult LV mass and concentric LVH. Several studies have shown that higher childhood BMI is associated with higher LV mass in childhood and adult life and an increased risk of adult LVH.[36 39 40]

There are few population-based data on the prevalence of echocardiographic abnormalities among South Asians. Indian migrants to the UK present with heart failure younger than white British men and women.[41] The echocardiographic imaging of healthy individuals in the UK London Life Sciences Prospective Population (LOLIPOP) study showed that South Asians have poorer diastolic function, threefold higher prevalence of LVH and a greater degree of concentric remodelling.[42] The UK Southall And Brent REvisited Study (SABRE) study showed that the impact of hyperglycaemia on LV mass and function was more adverse in South Asians.[15] One previous Indian study has investigated LV mass in relation to birth size; it found low mean LV mass compared with white Caucasian populations and an association of longer birth length with higher adult LV mass.[43 44]

Thus, several factors along the life course are related to adult LV hypertrophy and dysfunction, including greater adiposity, longer duration of adiposity, higher blood pressure and lipids, impaired glucose tolerance and early-life growth patterns (figure 2). The current IndEcho study was designed to investigate these relationships in two large population-based Indian birth cohorts.

## Hypotheses

► The prevalence of LVH and LV dysfunction and carotid intimal-media thickness (cIMT) will be positively related to current midlife cardiometabolic risk factors;

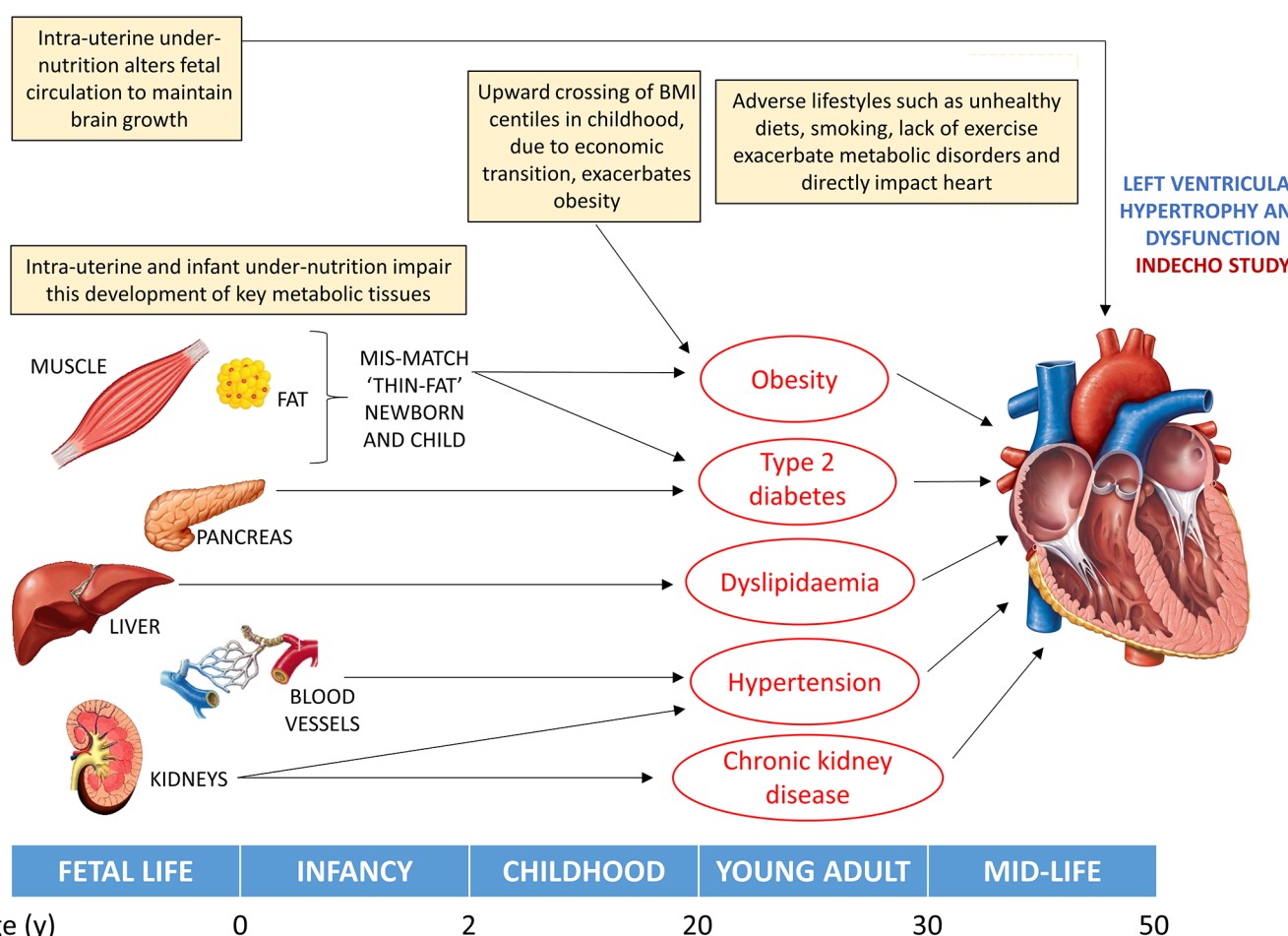

**Figure 2** Pathways to altered left ventricular (LV) structure and function that will be investigated in IndEcho. Factors contributing to LV hypertrophy. Intrauterine undernutrition alters the fetal circulation, which may have direct and persistent effects on ventricular structure. Intrauterine and infant undernutrition impairs the development of key metabolic tissues (muscle, pancreas, liver, blood vessels and kidneys) during critical periods of growth and increases adipose tissue deposition, leading to the 'thin-fat' phenotype and later obesity, T2DM, dyslipidaemia, hypertension and chronic renal disease, which adversely impact on LV size and function. BMI, body mass index; LV, left ventricular; T2DM, type 2 diabetes mellitus.

- ► After adjusting for current cardiometabolic risk factors, LVH, LV dysfunction and cIMT will be positively related to cardiometabolic risk factors measured 12–16 years ago in young adulthood;
- ► LVH, LV dysfunction and higher cIMT will be associated with lower birth weight, lower weight in infancy (the first two postnatal years) and faster BMI gain during childhood and adolescence;
- ► The associations of LVH, LV dysfunction and cIMT with adult cardiometabolic risk factors will be stronger in men and women who had lower birth or infant weight.

## METHODS AND ANALYSIS
### Study design
Multicentre observational cohort study.

### Study population
The IndEcho study started in November 2016 and will continue until 2019. Participants will be recruited from two Indian birth cohorts: the New Delhi Birth Cohort (NDBC)[29 45] and Vellore Birth Cohort (VBC).[46] Both were established in 1969–1973, originally to study maternal health and pregnancy outcomes. The participants were measured at various time points through infancy, childhood, adolescence and early adulthood to assess growth and were subsequently followed up at approximately 30 and 40 years of age to measure a range of CVD risk markers. Flow charts of the different stages of follow-up and various measurements are provided in figure 3. Currently, the participants are aged 43–50 years and new cases of glucose intolerance, T2DM and hypertension

are emerging alongside rapid transitions in lifestyle and socioeconomic status (SES). For the IndEcho study, we aim to recruit approximately 3000 individuals from both cohorts combined.

### The NDBC
In 1969–1972, 20 755 married women of reproductive age living in a 12 km² area of South Delhi were recruited (figure 3). The cohort included 8181 singleton live births from 9169 pregnancies among these women. Gestational age was derived from last menstrual period (LMP) dates. The birth weight, length and head circumference of the babies were recorded within 72 hours of birth (n=7119) and thereafter 3 monthly up to the age of 12 months (n=4104) and 6 monthly until the age of 21 years (n=2892). An average 23 sets of measurements were recorded for each individual from birth until 21 years. The first adult study took place in 1998–2002; 2584 (32%) of the original cohort were retraced, of whom 1526 men and women (then aged 28–31 years) participated in a study of CVD risk factors.[29] Data on their SES, attained education, family history of disease, tobacco and alcohol consumption, diet and physical activity were obtained using standardised questionnaires. Clinical measurements included anthropometry, blood pressure, an ECG and analysis of plasma glucose (during an oral glucose tolerance test), insulin, lipids and proinflammatory markers. In 2008 (at age 34–39 years), the same measurements were repeated in 1100 cohort members and additionally, cIMT and brachial artery endothelial function were measured[47] and body composition was assessed using dual-energy

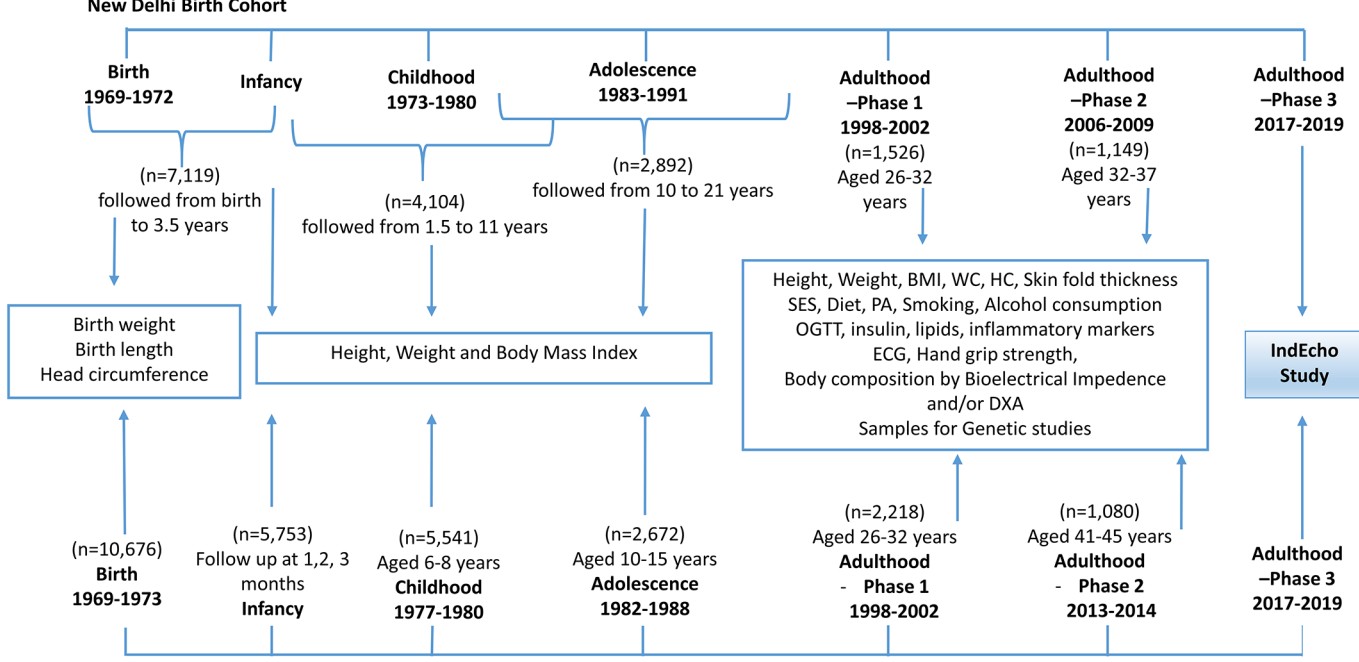

**Figure 3** Flow chart of various stages of follow-up of the New Delhi and Vellore Birth cohorts and measurements recorded at each stage. BMI, body mass index; DXA, dual energy X-ray absorptiometry; HC, hip circumference; OGTT, oral glucose tolerance test; PA, physical activity; SES, socioeconomic status.

X-ray absorptiometry (DXA) in a subset. Excluding 18 known deaths (from a total of 1526 who were followed up during 1998–2002), we aim to recruit as many as possible of the 1508 alive NDBC participants.

## The VBC

From 1969–1973, 20 626 women of reproductive age were recruited within defined areas of Vellore town and adjoining rural villages in Tamil Nadu, in South India[48] (figure 3). The areas of Vellore town were selected to represent different socioeconomic groups. The cohort comprised 10 691 singleton live babies born to the 20 626 women recruited. Weight and length were recorded at birth (n=10 676) and subsequently in infancy (in the first 3 months, n=5753), childhood (6–8 years, n=5541) and adolescence (10–15 years, n=2672). Gestational age was determined from the mother's LMP dates. Depending on available funding, VBC members had up to 3 measurements in the first 3 months, up to 2 measurements between 6 and 8 years and up to 5 measurements between 10 and 15 years. The first adult follow-up took place in 1998–2002. Cohort members for whom all birth measurements were available (n=4092) were retraced to participate in an adult follow-up during 1998–2002. Of these 2572 were contactable, and 2218 men and women, then aged 26–32 years, consented to participate in the study. CVD risk factors were assessed using a similar protocol to that for NDBC.[49] Subsequently in 2013–2014, 1080 participants (50.1% urban), aged 41–45 years, took part in a body composition study, in which the same anthropometric and biochemical parameters were recorded along with detailed body composition using DXA. Excluding 62 known deaths (from a total of 2218 who were followed up during 1998–2002), we aim to recruit as many as possible of the 2156 alive VBC members.

The total number of participants in phases 1 and 2 of adult follow-up in the NDBC and VBC and reasons for losses to follow-up are summarised in online supplementary table 1. We will minimise loss to follow-up through the following algorithm until all non-responders are accounted for: telephone calls/postal letters, meetings with neighbours by field workers and contact with local municipality/postal services to track addresses of migrated individuals.

## Measurements in the IndEcho study

Measurements will include anthropometry (height, weight, waist and hip circumferences, skinfold thickness), body composition by DXA (Vellore only) and bioelectrical impedance, hand grip strength, blood pressure, biochemical measurements (an oral glucose tolerance test, fasting plasma insulin and lipids and urinary albumin/creatinine ratio, as a measure of microalbuminuria). Details of the methods used for these measurements are given in table 1. Lifestyle factors (tobacco and alcohol consumption, diet, physical activity, occupation and SES) will be reassessed. Standard questionnaires for diet (food frequency questionnaire),[50] SES (Standard of Living Index),[51–53] Global

| Table 1 | IndEcho study procedures and platforms used |
| --- | --- |
| **Study-related procedures** | **Methods/platforms used** |
| Questionnaire assessments | |
| Diet | FFQ |
| Physical activity | GPAQ |
| Socioeconomic status | NFHS Standard of Living Index |
| Smoking | NFHS-2 Household Questionnaire |
| Alcohol consumption | NFHS-2 Household Questionnaire |
| Anthropometry | |
| Height | Stadiometer |
| Weight | Digital weighing scales |
| Waist circumference | Non-stretchable tape |
| Hip circumference | Non-stretchable tape |
| Blood pressure | Omron M3 |
| Biochemistry | |
| Glucose—fasting | Enzymatic method (autoanalyser) |
| Glucose—120 min | Enzymatic method (autoanalyser) |
| Insulin—fasting | Vellore: Radiofluorimetric method |
| Cholesterol | Delhi: Chemiluminescence immunoassay CHOD-PAP Enzymatic colorimetric method |
| Triglycerides | GPO-PAP Enzymatic colorimetric method |
| HDL cholesterol | Direct—two-step enzymatic |
| LDL cholesterol | Direct—enzymatic colorimetric method |
| Urinary ACR | Jaffe Method |
| Skinfold thickness | John Bull/Harpenden skinfold calliper |
| Bioimpedence | Tanita BC-418/Bodystat 2500 |
| Hand grip | JAMAR dynamometer |
| DXA | Hologic Discovery |
| ECG | |
| Echocardiography | Philips CX50 Compact Xtreme system |
| cIMT | Philips CX50 Compact Xtreme system |

FFQ developed by the National Institute of Nutrition, Hyderabad.[51]
SES Questionnaire developed by NFHS-2, 1998–1999.[52]
GPAQ developed by WHO.[53]
ACR, albumin-creatinine ratio; cIMT, carotid intimal-media thickness; DXA, dual energy X-ray absorptiometry; FFQ, Food Frequency Questionnaire; GPAQ, Global Physical Activity Questionnaire; HDL, high-density lipoprotein; LDL, low-density lipoprotein; NFHS, National Family Health Survey; SES, socioeconomic status.

Physical Activity Questionnaire,[54 55] smoking and alcohol consumption[56] will be used. Details on the key covariates relating to life style are summarised in online supplementary table 2.

Impaired glucose tolerance, impaired fasting glucose and T2DM will be diagnosed based on fasting glucose concentration and glucose concentration 120 min after a 75 g oral glucose load, using WHO criteria.[57] Hypertension will be defined as systolic pressure >140 mm Hg or diastolic pressure >90 mm Hg or on treatment for hypertension.[58] We will use International Diabetes Federation criteria for the following outcomes[59]: overweight and obesity: BMI >25 and >30 kg/m[2,] respectively; central obesity: a waist circumference >90 cm in men and ≥80 cm in women; hypertriglyceridaemia: plasma triglyceride concentration >1.7 mmol/L and low HDL cholesterol:<1.03 mmol/L in men and <1.29 mmol/L in women.

### Echocardiography

Cardiac chamber dimensions and systolic and diastolic function will be assessed using transthoracic echocardiography. All measurements will be performed according to the American and European Societies for Echocardiography guidelines for chamber quantification[60 61] and left ventricular diastolic function.[62] The same machine (Philips CX50 Compact Xtreme, Bothell, USA) and transducer (C5-1 Purewave curved array transducer) will be used in both centres and images will be analysed using the Freeland digitiser and software packages (Alpharetta, USA). M-mode and two-dimensional (2D) echocardiography in the parasternal long axis, midpapillary short axis and apical two-chamber and four-chamber views will be used to measure relative wall thickness (RWT; an index of wall thickness relative to internal dimensions). LVH will be defined based on any one of the following criteria: (1) LV mass >150 g in women and >200 g in men; (2) posterior wall thickness >1.1 cm or (3) LV mass indexed to body surface area (BSA) >95 g/m$^2$ in women and 115 g/m$^2$ in men.[61] Concentric LVH, which is an independent CVD risk factor, particularly in hypertensive patients, will be defined as increased RWT >0.42 in the presence of normal LV mass. LV systolic dysfunction will be assessed using measurements of fractional shortening, ejection fraction and global longitudinal strain (GLS). Ejection fraction will be measured using Simpson's biplane method in apical 2 and 4 chamber views. GLS will be calculated offline using speckle-tracking technology from the acquired 2D; two-chamber, three-chamber and four-chamber views using the in-built QLAB software in the CX50 echo machine. LV diastolic dysfunction will be assessed using mitral valve inflow velocities, mitral annular tissue Doppler velocities, left atrial volume index and tricuspid regurgitation jet velocity as per the guidelines.[62]

Scans will be performed by experienced echo technicians, two per centre, after a joint 2-day protocol-specific training session. Following this, the technical quality of scans will be checked by the senior cardiologist locally in each centre, and a set of 10 scans will be exchanged between centres for an independent quality assessment. Scans will be read by experienced cardiologists in each centre (AR in New Delhi and VTS and AGA in Vellore). A random 10% of all scans recorded in the study will be exchanged and read by the other centre to assess interobserver variability.

### cIMT measurement

Measurements of cIMT will be made bilaterally using high-resolution B-mode ultrasound with the same machine (Philips CX50 Compact Xtreme) and a 10-MHz linear array probe (Philips) and quantified using the CX50-IMT Philips Quantification application according to American Society for Echocardiography guidelines.[63] Each common carotid artery will be imaged in three different projections (anterior, lateral and posterior) proximal to the bifurcation. Measurements of IMT will be taken at a plaque-free zone in the far wall of the common carotid artery, captured at the end of diastole. The mean of three readings will be taken. The presence or absence of plaque, defined as a focal structure that encroaches on the arterial lumen by at least 0.5 mm or is more than 50% of the surrounding IMT or has a thickness greater than or equal to 1.5 mm will also be recorded. Between measurer reproducibility in the LV parameters and cIMT will be assessed by the exchange of 10% of scans from each site.

### ECG

All participants will undergo a 12-lead ECG recording. ECGs will be reviewed by a cardiologist, and if there are changes suggestive of ischaemia (ST-depression and/or T wave inversion), detailed Minnesota Coding will be performed. We will use Minnesota codes 1–1 and 1–2, 4–1 and 4–2, 5–1 and 5–2 and 7–1 to indicate ischaemic heart disease.[64]

### Body composition assessments

Body composition (total lean mass, fat mass and fat per cent) will be assessed using bioelectrical impedance in both centres (Tanita BC-418 in NDBC and Bodystat-2500 in VBC). In VBC, a DXA examination will be additionally performed (Hologic Discovery) to obtain total and depot specific lean mass, fat mass and total fat percentage. DXA body composition analysis was assessed among NDBC participants in 2009[65] and will not be re-examined in IndEcho.

A copy of the full protocol can be obtained from the corresponding author.

### Sample size and analysis

We will target the surviving cohort members (1508 from NDBC and 2156 from VBC) who participated in follow-ups between 1998 and 2002. From initial tracing in early 2016, a total of 3500 individuals were recontacted, among whom we expect approximately 1250 in Delhi and 1750 in Vellore to participate (total 3000) in the current protocol.

Exposures in early life will include weight and length at birth: weight, height and BMI and independent

**Table 2** Comparison of early growth measurements of studied and not-studied participants during phase 1 adult follow-up

| Measurement | New Delhi birth cohort | | | | Vellore birth cohort | | | |
| | Male | | Female | | Male | | Female | |
| | Studied as adult | | Studied as adult | | Studied as adult | | Studied as adult | |
| | Yes | No | Yes | No | Yes | No | Yes | No |
|---|---|---|---|---|---|---|---|---|
| **Birth** | | | | | | | | |
| n | 808 | 2642 | 569 | 2497 | 1159 | 1712 | 1058 | 1741 |
| Weight (kg) | 2.89±0.44 | 2.86±0.46 | 2.79±0.38 | 2.78±0.44 | 2.85±0.53 | 2.83±0.61 | 2.78±0.5 | 2.76±0.54 |
| Length (cm) | 48.8±2.2 | 48.7±2.4 | 48.3±1.9 | 48.1±2.2 | 48.3±3.0 | 47.8±4.6 | 47.8±3.0 | 47.3±4.3 |
| Ponderal index (kg/m$^3$) | 24.8±2.6 | 24.7±2.9 | 24.7±2.5 | 24.9±2.9 | 25.7±7.6 | 25.6±7.3 | 25.7±6.6 | 25.9±7.3 |
| Gestation (week) | 39.2±2.2 | 39.2±2.3 | 39.6±2.0 | 39.6±2.2 | 38.2±2.8 | 38.1±2.9 | 38.3±2.8 | 38.4±2.8 |
| **Infancy (3 months)** | | | | | | | | |
| n | 660 | 2140 | 503 | 2105 | 845 | 1011 | 791 | 1108 |
| Weight (kg) | 5.50±0.7 | 5.46±0.8 | 4.96±0.7 | 4.99±0.7 | 4.28±0.8 | 4.17±0.9 | 4.04±0.7 | 3.91±0.8 |
| Height (cm) | 59.6±2.4 | 59.5±2.7 | 58.0±2.4 | 58.1±2.5 | 55.2±3.1 | 55.1±3.6 | 54.3±2.9 | 54.2±3.2 |
| BMI (kg/m$^2$) | 15.4±1.5 | 15.4±1.6 | 14.7±1.6 | 14.7±1.6 | 14.1±1.9 | 13.6±2.2 | 13.6±1.7 | 13.12±1.92 |
| **Childhood (6 years)** | | | | | | | | |
| n | 837 | 1092 | 608 | 1219 | 1001 | 601 | 933 | 645 |
| Weight (kg) | 17.2±2.2 | 17.3±2.3 | 16.4±2.0 | 16.5±2.2 | 15.8±4.4 | 15.5±4.4 | 15.4±4.3 | 15.4±4.2 |
| Height (cm) | 108.4±5.2 | 108.9±5.5 | 107.0±5.1 | 107.2±5.5 | 102.4±18.7 | 100.1±26.4 | 100.6±19.7 | 100.6±23.5 |
| BMI (kg/m$^2$) | 14.6±1.1 | 14.5±1.1 | 14.3±1.1 | 14.3±1.2 | 15.4±4.4 | 15.3±4.1 | 15.6±4.8 | 15.2±4.11 |
| **Adolescence (15 years)** | | | | | | | | |
| n | 616 | 435 | 481 | 592 | 775 | 467 | 727 | 508 |
| Weight (kg) | 44.6±9.2 | 45.8±9.6 | 44.6±7.8 | 44.2±7.5 | 27.7±5.8 | 27.5±6.3 | 28±5.37 | 28.5±6.1 |
| Height (cm) | 159.2±8.7 | 160.2±8.4 | 153.2±5.7 | 153.3±6.2 | 136.5±9.9 | 137.1±13.6 | 135.3±8.0 | 136.5±9.9 |
| BMI (kg/m$^2$) | 17.5±2.5 | 17.7±2.7 | 18.9±3.0 | 18.8±2.7 | 14.7±1.4 | 14.6±1.6 | 15.2±1.9 | 15.2±1.8 |
| **Adulthood** | | | | | | | | |
| n | 886 | — | 640 | — | 1159 | — | 1053 | |
| Age (years) | 29.2±1.3 | — | 29.2±1.4 | — | 27.9±1.1 | — | 28.3±1.2 | — |
| Weight (kg) | 71.8±14.0 | — | 59.2±13.4 | — | 57.4±11.4 | — | 49.3±10.6 | — |
| Height (cm) | 169.7±6.4 | — | 154.9±5.7 | — | 166.4±6.7 | — | 153.8±6.0 | — |
| BMI (kg/m$^2$) | 24.9±4.3 | — | 24.6±5.1 | — | 20.7±3.5 | — | 20.8±4.1 | — |

BMI, body mass index.

conditional weight and height estimates of growth during infancy, childhood and adolescence, as previously described.[48] Because early growth measurements were measured at different time points and at slightly different ages in both cohorts, and owing to the skewness of BMI, the data will be transformed into SD scores using the lambda-mu-sigma (LMS) method.[66] The main analysis will use all available data at each time point; it will also be repeated using the subset of participants having data for all time points (birth, infant, childhood and adolescent measurements). Exposures in young adult life will include lifestyle characteristics (diet, physical activity, smoking and alcohol consumption, SES, urban/rural residence and history of rural–urban migration (Vellore only)), anthropometry, lean and fat mass measured by DXA and bioimpedance, metabolic risk markers (plasma glucose, insulin and lipids) and categorical abnormalities as defined above.

We will check the distributions of all variables and perform appropriate transformations. We will explore relationships between LV wall thickness and mass (as continuous variables) with current body size and make appropriate adjustments, including the conventional adjustment for body surface area and height.[60] We will create categorical variables to represent LVH and ventricular dysfunction based on standard cut-offs for LV mass as described above.[61] We will examine associations of early life and young adult exposures with outcomes, using

linear and logistic regression as appropriate, with and without adjustment for potential confounders and covariates (including age, sex, history of migration, current body size and composition). We will also use the conditional growth modelling approach to relate growth in BMI and height during discrete age periods with adult outcomes.[67 68] We will explore interactions between early life and young adult exposures and between early life exposures and current measures, using product terms. The role of young adult lifestyle and CVD risk markers as mediators of associations between early-life exposures and outcomes will also be examined. To test the representativeness of the IndEcho participants and assess the risk of bias, we will compare their early growth variables (birth length, birth weight and height, weight and BMI in infancy, childhood and early adulthood) with members of the original cohorts who did not participate as we have done previously (table 2). Since all analyses in the IndEcho study will be 'internal' (within the sample of participants) bias will occur only if the associations between early growth and adult outcomes differ between those who participate in IndEcho and those who were not studied. We cannot think of a mechanism by which such differences could occur. We will examine the data from men and women, from Delhi and Vellore and (in Vellore) among rural and urban participants separately and assess heterogeneity of associations and scope for pooling.

Using tests at 5% significance and a total sample size of 3000, we will have 80% power to detect an association of 0.05 SDs of a continuous outcome (eg, LV mass) per SD change in a continuous predictor (eg, birth weight or conditional weight gain in infancy). For the Vellore cohort alone (based on n=1750), this figure is 0.07 SDs and for the Delhi cohort alone (based on n=1250), it is 0.08 SDs. These figures compare favourably with the association shown between infant weight and adult LV mass in the Hertfordshire cohort (0.17 SD per SD change in infant weight).[26] For a binary outcome such as LV diastolic dysfunction (Grades 1 and 2), assuming 10% prevalence and a test at 5% significance level, we will have 80% power to detect an association of 0.17 log odds per SD change in a continuous predictor, equivalent to an OR of 1.19 (Vellore 1.25, Delhi 1.30).

## DISCUSSION

To the best of our knowledge, IndEcho will be the first study among South Asians to provide data linking birth size and childhood growth, as well as prior measures of CVD risk markers and lifestyle factors such as physical activity and smoking in young adulthood, with cardiac structure and function and in middle age. It will provide an opportunity to study interactions between early life, young adult and concurrent risk factors in relation to myocardial outcomes.

The main strengths of the study are that it includes participants from two large Indian birth cohorts that have serial childhood growth data, measured prospectively by trained research staff and risk factors for CVD (including lifestyle risk factors) assessed 16–20 years ago. Harmonised methods will be used in both cohorts. The participants represent north and south Indian populations, different socioeconomic strata and rural and urban settings. Because we have serial measures of SES, collected using identical methods, IndEcho provides an opportunity to assess the impact of economic transition on cardiac health in Indians. At this age, we will detect subclinical cardiac disease in an apparently healthy population where obesity, hypertension and T2DM are escalating. In 1998–2002 at age 26–32 years, the prevalence of obesity (BMI >30 kg/m$^2$), hypertension (systolic >140 mm Hg or diastolic >90 mm Hg or on treatment for hypertension) and diabetes (WHO criteria) was 11%, 8% and 4%, respectively, in NDBC[29] and 2%, 3% and 3% in VBC.[31] Approximately 5 years later, at 34–39 years, the prevalence had more than doubled in NDBC (23%, 26% and 9%).[45] In the 2013–2014 VBC follow-up, the equivalent data (16%, 20% and 17%) showed a more than fivefold increase in prevalence in just over a decade (unpublished data).

Like all birth cohorts of this age, an important limitation is loss to follow-up, mainly due to deaths in childhood and migration out of the original study area. However, current cohort members were similar in early life to the rest of the original cohorts (table 2).[29 48] To create bias, the relationship between the predictors (eg, birth weight or young adult diabetes) and outcomes (eg, LV mass) would have to differ between those studied and not studied. Another limitation is that, because this is a field-based study, echocardiography will be used rather than cardiac MRI which is the current gold standard method for measuring myocardial structure.

The findings of the IndEcho study will help plan effective strategies in early life and/or young adulthood to prevent adult CVD in India and migrant South Asian populations. We expect that it will identify periods during early life when nutritional interventions may prevent later cardiac disease. As the prospective follow-up continues in future and as cohort members start to develop cardiac disease, we will also be able to define the value of LV measurements for predicting cardiovascular morbidity and mortality, which will be the first such data from India.

## ETHICS AND DISSEMINATION

IndEcho does not involve any invasive or risky procedures for participants. Participant information sheets and consent forms are made available in English and local languages, describing the purpose of the study and the various procedures in lay terms. Participation is voluntary and participants provide written informed consent before enrolment. Findings from the study will be disseminated through presentations at both international and national conferences and results generated will be published in international peer-reviewed journals.

**Author affiliations**
[1]MRC Lifecourse Epidemiology Unit, University of Southampton, Southampton General Hospital, Southampton, UK
[2]Oxford Center for Diabetes, Endocrinology and Metabolism, Radcliffe Department of Medicine, University of Oxford, Oxford, UK
[3]Centre for Chronic Disease Control, New Delhi, India
[4]Department of Cardiology, All-India Institute of Medical Sciences, New Delhi, India
[5]Departments of Cardiology, Biostatistics, Endocrinology and Clinical Biochemistry, Christian Medical College, Vellore, Tamil Nadu, India
[6]Department of Paediatrics, Sunder Lal Jain Hospital, New Delhi, India
[7]Department of Paediatrics, Sitaram Bhartia Institute of Science and Research, New Delhi, India
[8]Public Health Foundation of India, New Delhi, India

**Correction notice** This article has been corrected since it first published. The Open access licence has been changed from CC BY-NC to CC BY.

**Acknowledgements** We thank the members of the NBDC and VBC and their families for their participation in research over more than four decades. We acknowledge the IndEcho field teams in Delhi and Vellore, which include administrative staff, field workers and supervisors, research officers, research nurses, phlebotomists, DXA and echo technicians, data management teams and data entry operators.

**Contributors** SKV, AR, VTS, BA, SKB, AAG, CO, FSK, HS, LR, NT, NJT, TVP, DP and CHDF conceived the study and wrote the study protocol. BS, FSG, KA, PSP, PA, SFXP and SS will contribute significantly to the acquisition of the data. CO, SKV, FK, DP, SKB, HS, AR, VTS, AGA, BA, PSP, SFXP and CHDF significantly contributed to the planning of analyses of the data. SKV and CHDF drafted the first version of the manuscript and all authors revised the manuscript critically for important intellectual content. All authors reviewed and approved the final manuscript and agreed to be accountable for all aspects of the work.

**Funding** The original cohort studies were supported by the National Center for Health Statistics, USA and the Indian Council of Medical Research. The two earlier follow-up studies in young adult life were supported by the British Heart Foundation. The IndEcho study is supported by British Heart Foundation Clinical Research Grant, no. CRM: 0022324. Professor Fall's work on the study is supported by the UK Medical Research Council (MRC) and the UK Department for International Development (DFID) under the MRC/DFID Concordat.

**Competing interests** None declared.

**Patient consent** Obtained.

**Ethics approval** The study has been approved by the research ethics committees of Sunder Lal JainHospital, New Delhi (13th August 2015; SLJ/IEC/1); Sitaram Bhartia Institute of Science and Research, New Delhi (23 October 2015; IEC/SBSR/2015/1); Centre for Chronic Disease Control, New Delhi (no.50/7/TF-CVD/15-NCD-II); All-India Institute of Medical Sciences, New Delhi (21 October 2015, IEC/NP-410/09.10.2015); Christian Medical College, Vellore (22 July 2015; IRB 9548 (OBSERV)) and the Faculty of Medicine, University of Southampton, UK (11 April 2016; RE 18694).

**Provenance and peer review** Not commissioned; externally peer reviewed.

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
