## [Reviewer comments · BMJ Open]

ARTICLE DETAILS

TITLE (PROVISIONAL)	The IndEcho study: Cohort study investigating birth size, childhood growth and young adult cardiovascular risk factors as predictors of mid-life myocardial structure and function in South Asians
AUTHORS	Vasan, Senthil; Roy, Ambuj; Samuel, Viji; Antonisamy, Belavendra; Bhargava, Santosh; Alex, Anoop; Singh, Bhaskar; Osmond, Clive; Geethanjali, Finney; Karpe, Fredrik; Sachdev, Harshpal; Agrawal, Kanhaiya; Ramakrishnan, Lakshmy; Tandon, Nikhil; THOMAS, NIHAL; Premkumar, Prasanna; Asaithambi, Prathepa; Princy, Sneha; Sinha, Sikha; PAUL, THOMAS; Prabhakaran, Dorairaj; Fall, Caroline

VERSION 1 – REVIEW

REVIEWER	Jingyan Tian Shanghai Institute of Endocrine and metabolic Diseases
REVIEW RETURNED	02-Oct-2017

GENERAL COMMENTS	Comments to the Authors: Vasan and colleagues propose to study approximately 3,000 men and women aged 43-50 years from two birth cohorts established in 1969-1973. The study will provide the first longitudinal data linking size at birth, childhood growth and cardiovascular disease risk markers in young adult life with myocardial structure and function in mid-life in South Asians The research design is well-organized and the methods were outlined and explained in details. The study is both interesting and of importance to understand birth size, childhood growth and young adult cardiovascular risk factors as predictors of mid-life myocardial structure and function in South Asians. Minor comments: In the New Delhi Birth Cohort (NDBC), the first adult study took place in 1998-2002; 1,526 men and women (then aged 28-31 years) participated in a study of CVD risk factors. In 2008 (at age 34-39 years) the same measurements were repeated in 1,149 cohort members. In the Vellore Birth Cohort (VBC), during 1998-2002. 2,218 men and women consented to participate in the study. Subsequently in 2013-14, 1,080 participants (50.1% urban), aged 41-45 years took part in
---

	a body composition study. About 50.1% participants took part in the 2013 study. What is the reason for missing data? Analysis of the reasons for missing data and taking measures to prevent it will increase the response rate of the IndEcho study.
--	--

REVIEWER	David Burgner Murdoch Children's Research Institute, Melbourne, Australia
REVIEW RETURNED	09-Oct-2017

GENERAL COMMENTS	This is an interesting and novel study from well-respected researchers that should provide important findings. It is generally well written and easy to follow. There are a few grammatical errors that warrant correction. The main issue is that the study doesn't quite 'hang together' as presented. It is somewhat unclear how the childhood growth and body composition parameters, cardio-metabolic risk factors and vascular phenotypes in early adulthood, and the echo outcomes that are proposed will actually be analysed. Are the young adult outcomes considered an outcome following the childhood exposures (and are there data to suggest an association?), or are they an exposure for the IndEcho outcomes, or both? I think this wealth of data makes the actual analysis potentially quite complex, which is why a statistical review might be helpful. Some issues for the authors to consider:  - In the abstract please specify what is meant by 'Target N' for the NDBC. Please give the actual number recruited at enrolment. - please specify the number by each sex and the number in each cohort retained at age 30y (including number of deaths/migration), as well as the estimated retention at the planned CV assessment - abstract line 12/13 and page 6 line 43: data is a plural noun - page 4 lines 50/51: "(muscle-thin but centrally adipose)". I am not sure this is common terminology - suggest rephrase - page 6 line 16. Unclear what 'plane of nutrition' refers to - suggest rephrase - please explain the term concentric LVH for non-experts and the significance - the hypotheses are not entirely clear and might benefit from rewriting; they don't fit with the title particularly clearly. One issue is that there are two adult measures of traditional CV risk factors and these are likely to be strongly correlated. It may be worth considering presenting the study as (i) an initial cross-sectional analysis at time of IndEcho assessment (ii) longitudinal analysis from early adulthood to echo outcomes (iii) childhood exposures and adult outcomes (as already done). It is also unclear how cIMT fits into the causal pathway. It is not mentioned in the abstract or in the hypotheses. It is clearly an interesting measure, as is endothelial function, to which the same issues are relevant. - inevitably there is considerable attrition in both cohorts. What steps will be taken to investigate how representative the IndEcho study participants are? - it would be useful to have more details of some of the key co-variates that are often hard to measure, including PA, SES etc. Consider summarising the exposure and outcome measures (and where appropriate their reproducibility) in an expanded table.
---

	 - It is unclear how LVH is defined? - will outcome measures such as LVH be considered as continuous variables, as well as using accepted cut-offs? this is often used for measures such as cIMT - will mean cIMT and/or maximum cIMT be measured? And how? (edge detection software, calipers etc?). Will cIMT be measured at end-diastole on ECG? - please define the parameters of ischaemia on ECG - it would be helpful to have some details of the childhood measures in the text and Figures 1 and 2 should be referenced in the text. Figure 3 was rather blurred and a little difficult to read (but clearly essential). - some discussion of how the BMI trajectories (Fig 1) will be analysed with respect to the early adult and IndEcho outcomes would be useful - endothelial function is not mentioned in Table 1. Reproducibility estimates for BP, CIMT and endothelial function would be useful - Table 1 could be expanded to include these for measures as appropriate - would the authors consider a Table of the characteristics of the participants at enrolment, in childhood and at young adulthood? - how representative of the Indian population are the cohorts? What parameters is this based upon? Is the cohort a truly representative? - please provide more details of how the study allows investigation of the impact of economic transition. Which analysis specifically addresses this? - with respect to retention and potential bias, it would be useful to have the key characteristics of the enrolment, childhood and young adult participants in a Table (as suggested previously)
--	--

REVIEWER	Olli Saarela University of Toronto, Canada
REVIEW RETURNED	01-Dec-2017

GENERAL COMMENTS	The manuscript summarizes a protocol for a birth cohort study of risk factors, in particular lower birth and infant weight, and childhood weight gain, of left ventricular hypertrophy and dysfunction later in life in India. There is evidence that low birth weight followed by weight gain in adolescence is associated with several cardiovascular disease risk factors in adulthood. However, there is less evidence of associations with cardiac structure and function, in particular in South Asians, which serves as the motivation for the planned study. The study is based on two existing Indian birth cohorts, the The New Delhi Birth Cohort and the The Vellore Birth Cohort. As part of the planned study, a further round of measurements will be collected on expected 3,000 members of the cohorts with earlier follow-up measurements available. The cohort members are now between 43-50 years old. The data to be collected include the cardiac structure outcome measurements using echocardiography. Power calculations for the planned association analyses are presented. Linear and logistic regression and conditional growth models are planned for analysis. Substantial attrition of the original birth cohorts, mainly due to death and migration, is mentioned as a limitation of the study. However, it is noted that the the remaining cohort members are similar to the original cohorts in terms of their early life characteristics. The study is funded and has ethics approval. I have just one question: is there evidence for the suggested participation rate (3,000 of the 3,500 re-contacted)?
--

VERSION 1 – AUTHOR RESPONSE

Editorial Requirements:

- Please revise the Strengths and Limitations section (after the abstract) to focus on the methodological strengths and limitations of your study. This section should not summarize the results.

Authors' response: We have now revised the strengths and limitations section as suggested by the editor.

Revision (Page 3):

- The strengths include the longitudinal study design with early growth data, which allows examination of the relationship of size at birth, childhood growth and cardiovascular disease risk markers in young adult life with myocardial structure and function in mid-life in South Asians.
- The two cohorts represent both rural and urban populations, north and south India, and diverse socio-economic strata.
- The study will use harmonised methods to measure anthropometry, biochemical risk factors and to characterise lifestyle factors.
- A limitation of the study is attrition of the cohorts due to deaths and migration (mainly in childhood).
- Because this is a field-based study, echocardiography will be used, rather than cardiac MRI, which is the current gold standard method for measuring myocardial structure.

Comments to the Authors:

Vasan and colleagues propose to study approximately 3,000 men and women aged 43-50 years from two birth cohorts established in 1969-1973. The study will provide the first longitudinal data linking size at birth, childhood growth and cardiovascular disease risk markers in young adult life with myocardial structure and function in mid-life in South Asians. The research design is well-organized and the methods were outlined and explained in details. The study is both interesting and of importance to understand birth size, childhood growth and young adult cardiovascular risk factors as predictors of mid-life myocardial structure and function in South Asians.

Minor comments:

In the New Delhi Birth Cohort (NDBC), the first adult study took place in 1998-2002; 1,526 men and women (then aged 28-31 years) participated in a study of CVD risk factors. In 2008 (at age 34-39 years) the same measurements were repeated in 1,149 cohort members. In the Vellore Birth Cohort (VBC), during 1998-2002. 2,218 men and women consented to participate in the study. Subsequently in 2013-14, 1,080 participants (50.1% urban), aged 41-45 years took part in a body composition study. About 50.1% participants took part in the 2013 study. What is the reason for missing data? Analysis of the reasons for missing data and taking measures to prevent it will increase the response rate of the IndEcho study.

Authors' response: Reasons for missing data in both cohorts between Phase-1 and Phase-2 of adult follow-up are summarized in the table below. We have now included this as Supplementary table 1 in the revised manuscript.

We apologise for the typo in the NDBC participation figures during Phase-2. The total participants in phase-2 were 1,100, not 1,149 (Ref47: Khalil et al. Int J Cardiol 2013; 167(4):1322-8). The changes have now been included in the revised manuscript Page 7, line 202.

The current IndEcho recruitment is aimed to include the maximum number of cohort members who participated in the Phase 1 adult follow-up. We will minimize loss to follow-up through the following algorithm, until all non-responders are accounted for: telephone calls/postal letters, meetings with neighbours by field workers, and contact with local municipality/postal services to track addresses of migrated individuals. We have included the above information in the revised manuscript Page 8, lines 232-237.

The expected recruitment is included in the revised manuscript as below:

Page 7, line 205-207: Excluding 18 known deaths (from a total of 1,526 who were followed up during 1998-2002), we aim to recruit as many as possible of the 1,508 alive NDBC participants.

Page 8, line 228-230: Excluding 62 known deaths (from a total of 2,218 who were followed up during 1998-2002), we aim to recruit as many as possible of the 2,156 alive VBC members.

Page 11, line 333-334: We will target the surviving cohort members (1,508 from NDBC and 2,156 from VBC) who participated in follow-ups between 1998 and 2002.

Reviewer: 2

Reviewer Name: David Burgner

Institution and Country: Murdoch Children's Research Institute, Melbourne, Australia

Please state any competing interests: none declared

Please leave your comments for the authors below

This is an interesting and novel study from well-respected researchers that should provide important findings. It is generally well written and easy to follow. There are a few grammatical errors that warrant correction.

The main issue is that the study doesn't quite 'hang together' as presented. It is somewhat unclear how the childhood growth and body composition parameters, cardio-metabolic risk factors and vascular phenotypes in early adulthood, and the echo outcomes that are proposed will actually be analysed. Are the young adult outcomes considered an outcome following the childhood exposures (and are there data to suggest an association?), or are they an exposure for the IndEcho outcomes, or both? I think this wealth of data makes the actual analysis potentially quite complex, which is why a statistical review might be helpful.

Some issues for the authors to consider:

- In the abstract please specify what is meant by 'Target N' for the NDBC. Please give the actual number recruited at enrolment.

Authors' response: We apologise for the lack of clarity. The 'target N' refers to the number of individuals who will be contacted for participation in the current IndEcho study. These will include all cohort members retained at age 30 years, excluding known deaths since then (18 from 1,526 in New Delhi and 62 from 2,218 in Vellore). We have now removed the term 'target' and revised the abstract accordingly.

Revision in manuscript (Abstract Page 2, lines 51-53): We propose to study approximately 3,000 men and women aged 43-50 years from two birth cohorts established in 1969-1973: the New Delhi Birth Cohort (NDBC; N=1,508) and Vellore Birth Cohort (VBC; N=2,156).

- please specify the number by each sex and the number in each cohort retained at age 30y (including number of deaths/migration), as well as the estimated retention at the planned CV assessment

Authors' response: Please refer to the table included in our response to Reviewer 1, above. This table, describing reasons for losses to follow-up, is now added as a Supplementary Table 1 in the revised manuscript.

- abstract line 12/13 and page 6 line 43: data is a plural noun

Authors' response: This has been edited now as "there are few data on their relationship to LV function" in the abstract section (Page 2, line 46 and Page 5, line 133 in the revised manuscript).

- page 4 lines 50/51: "(muscle-thin but centrally adipose)". I am not sure this is common terminology - suggest rephrase

Authors' response: We have now changed this to "low muscle mass". The edited line in the revised version (Page 4, line 104) reads as "a 'thin-fat' body composition (low muscle mass but centrally adipose)"

- page 6 line 16. Unclear what 'plane of nutrition' refers to - suggest rephrase

Authors' response: We have now removed the term 'plane of nutrition' and the edited line (Page 5, line 116-117) reads as "The risks associated with lower birth weight appear to be increased on subsequent exposure to greater child or adult weight gain."

- please explain the term concentric LVH for non-experts and the significance

Authors' response: Concentric LVH is an independent CVD risk factor, particularly in hypertensive patients, and is defined as increased relative wall thickness (RWT) in the presence of normal LV mass. A relative wall thickness (RWT) >0.42 is the recommended cut-off to define concentric LVH. The description of concentric LVH is now included in the revised manuscript (Pages 9 & 10, Lines 277-279).

- the hypotheses are not entirely clear and might benefit from rewriting; they don't fit with the title particularly clearly. One issue is that there are two adult measures of traditional CV risk factors and these are likely to be strongly correlated. It may be worth considering presenting the study as (i) an initial cross-sectional analysis at time of IndEcho assessment (ii) longitudinal analysis from early adulthood to echo outcomes (iii) childhood exposures and adult outcomes (as already done). It is also unclear how cIMT fits into the causal pathway. It is not mentioned in the abstract or in the hypotheses. It is clearly an interesting measure, as is endothelial function, to which the same issues are relevant.

Authors' response: We agree with the reviewer that the hypotheses could be clearer, and have therefore changed them in the revised manuscript. The title remains unchanged. We agree that cIMT does not fit into the causal pathway, and this will be considered as an additional outcome.

Revised hypotheses (Page 6, Lines 151-164):

1. The prevalence of LVH and LV dysfunction, and cIMT, will be positively related to current mid-life cardio-metabolic risk factors
2. After adjusting for current cardio-metabolic risk factors, LVH, LV dysfunction and cIMT will be positively related to cardio-metabolic risk factors measured 12-16 years ago in young adulthood.
3. LVH, LV dysfunction and higher cIMT will be associated with lower birth weight, lower weight in infancy (the first two post-natal years) and faster BMI gain during childhood and adolescence.
4. The associations of LVH, LV dysfunction and cIMT with adult cardiometabolic risk factors will be stronger in men and women who had lower birth or infant weight.

- inevitably there is considerable attrition in both cohorts. What steps will be taken to investigate how representative the IndEcho study participants are?

Authors' response: We agree with the reviewer that there has been considerable attrition in both cohorts. This was mainly due to deaths and migration, predominantly during early childhood. Reasons for 'lost to follow-up' between adult Phase 1 and Phase 2 are included in Supplementary Table 1.

Much of the attrition in VBC was due to funding restrictions in Phase 2. We anticipate that many of these cohort members will be available for the IndEcho study.

In order to test the representativeness of the IndEcho participants, we will compare their early growth variables (birth length, birth weight, and height, weight and BMI in infancy, childhood and early adulthood) with members of the original cohorts who did not participate. We have done this in previous phases of adult follow-up (see table at the end of this document). We show the comparison for those studied and not studied in phase 1 of adult follow-up in both cohorts. This table is now also included in the revised manuscript (Table 2). Since all analyses in the IndEcho study will be 'internal' (within the sample of participants) bias will occur only if the associations between early growth and adult outcomes differ between those who participate in IndEcho and those who were not studied. We cannot think of a mechanism by which such differences could occur. We have included relevant text in the revised manuscript Page 12, lines 366-374.

- it would be useful to have more details of some of the key co-variables that are often hard to measure, including PA, SES etc. Consider summarising the exposure and outcome measures (and where appropriate their reproducibility) in an expanded table.

Authors' response: We have now summarized the key co-variables relating to lifestyle (SES, PA and smoking and alcohol consumption) in Supplementary table 2 of the revised manuscript. These will all be assessed in IndEcho using existing standard questionnaires that have been validated, either in Indian populations or other large population cohorts, as cited in the table. We have previously used the same set of questionnaires (except the GPAQ) in the 1998-2002 adult follow-up in both NDBC and VBC.

- It is unclear how LVH is defined?

Authors' response: LVH will be defined based on any one of the following criteria: (i) LV mass >150 g in women and >200 g in men, (ii) posterior wall thickness >1.1 cm or (iii) LV mass indexed to BSA >95 g/m² in women and 115 g/m² in men (Ref: Lang et al, J of Am Soc Echocardiogr. 2015). This is now included in the revised manuscript under Methods: Echocardiography section (Page 9, Lines 274-277).

- will outcome measures such as LVH be considered as continuous variables, as well as using accepted cut-offs? this is often used for measures such as cIMT

Authors' response: In data analysis, outcomes will be considered both as continuous and dichotomised/ categorized variables. For example, LV mass will be used as a continuous variable, and dichotomised according to the above cut-offs for LVH. We will use both linear and logistic regression analysis, adjusted for potential confounders. This is explained in detail in Page 12 (lines 353-357) under section sample size and analysis.

- will mean cIMT and/or maximum cIMT be measured? And how? (edge detection software, calipers etc?). Will cIMT be measured at end-diastole on ECG?

Authors' response: We have now added a more detailed explanation of the cIMT measurement in the revised manuscript (Page 10, lines 298-310).

cIMT will be measured using high-resolution B-mode ultrasonography and a 10-MHz linear array probe (Philips), and quantified using the CX50-IMT Philips Quantification application. Measurements of IMT will be taken at a plaque-free zone in the far wall of the common carotid artery, captured at the end of diastole. The mean of three readings will be taken. This information is now included in the revised manuscript

- please define the parameters of ischaemia on ECG

Authors' response: Ischaemic changes on ECG will be based on the appearance of ST-depression and/or T wave inversion. This has been edited in the revised manuscript (Page 11, Lines 314-318) as

– “ECGs will be reviewed by a cardiologist, and if there are changes suggestive of ischaemia (ST-depression and/or T wave inversion), detailed Minnesota Coding will be performed. We will use Minnesota codes 1-1 and 1-2, 4-1 and 4-2, 5-1 and 5-2 and 7-1 to indicate ischaemic heart disease.

- it would be helpful to have some details of the childhood measures in the text and Figures 1 and 2 should be referenced in the text. Figure 3 was rather blurred and a little difficult to read (but clearly essential).

Authors' response: The early growth measurements recorded from birth to adolescence are included in Figure 3 and all figures are referenced appropriately in the manuscript

Figure 1: referenced on Page 5, line 121;

Figure 2 referenced on Page 6, lines 145-146;

Figure 3 referenced on Page 7: lines 179 & 188 and Page 8, line 213.

We apologize for the poor image quality of Figure 3 and have now included a revised image.

- some discussion of how the BMI trajectories (Fig 1) will be analysed with respect to the early adult and IndEcho outcomes would be useful

Authors' response: As suggested, we have now added a detailed section on the analysis of BMI trajectories as below.

Revision in manuscript (Page 11, lines 340-345): Because early growth measurements were measured at different time points and at slightly different ages in both cohorts, and owing to the skewness of BMI, the data will be transformed into SD scores using the LMS method (Ref 66: Cole TJ, Green PJ, Smoothing reference centile curves: the LMS method and penalized likelihood. *Statistics in Medicine* 1992, 11: 1305-1319).

Revision in manuscript (Page 12, lines 361-362): We will also use the conditional growth modelling approach to relate growth in BMI and height during discrete age periods with adult outcomes (Refs: 67 & 68).

- endothelial function is not mentioned in Table 1. Reproducibility estimates for BP, CIMT and endothelial function would be useful - Table 1 could be expanded to include these for measures as appropriate

Authors' response: We are not assessing endothelial function in the IndEcho study. Between measurer reproducibility in the LV parameters and cIMT will be assessed by the exchange of 10% of scans from each site. This information is provided in Page 10, lines 309-310.

- would the authors consider a Table of the characteristics of the participants at enrolment, in childhood and at young adulthood?

Authors' response: We thank the reviewer for this suggestion. We have now included early life characteristics of the NDBC and VBC as Table 2 in the revised manuscript. We have referred to this table on page 12, line 370 and page 14, line 420 of the manuscript.

- how representative of the Indian population are the cohorts? What parameters is this based upon? Is the cohort a truly representative?

Authors' response: The Indian population is very geographically and socio-economically heterogeneous. Both cohorts were recruited from delineated geographical areas, and are representative of those areas, but cannot be considered representative of the whole Indian population. The VBC includes both rural and urban participants, and efforts were made to cover socio-economically diverse areas of the urban setting. The literacy rate in the VBC (80%) tallies with the figure from the 2011 Vellore census (79.7%; <http://www.censusindia.gov.in/2011census/>). The NDBC is based on a large area within a single city, and one with a predominantly middle-class population. It does not include any rural participants or participants from the very poor or very wealthy socio-economic strata. To obtain a truly representative sample in a country like India would require a very large survey. Our chief aim is not to provide 'normal data' for India, but to examine associations

within these two reasonably diverse cohorts between factors measured earlier in life and adult echocardiographic outcomes.

Revision in manuscript (page 13, lines 405): We have deleted the line in the Discussion section stating that the IndEcho participants are 'representative of the general Indian population'.

- please provide more details of how the study allows investigation of the impact of economic transition. Which analysis specifically addresses this?

Authors' response: The IndEcho study is not designed specifically to investigate the impact of economic transition but does provide some opportunities. It will use the same standard of living index (SLI) questionnaire used previously in the 1998-2002 adult follow-up. Using data from both phases of follow-up, we will estimate how initial SLI and change in SLI predict outcomes.

Revision in manuscript (page 13, lines 405-407): We have clarified this point in the Discussion section.

- with respect to retention and potential bias, it would be useful to have the key characteristics of the enrolment, childhood and young adult participants in a Table (as suggested previously)

Authors' response: We have now included a table containing information on key early growth characteristics from birth to young adulthood (Table 2 in the revised manuscript).

Reviewer: 3

Reviewer Name: Olli Saarela

Institution and Country: University of Toronto, Canada

Please state any competing interests: None declared

Please leave your comments for the authors below

The manuscript summarizes a protocol for a birth cohort study of risk factors, in particular lower birth and infant weight, and childhood weight gain, of left ventricular hypertrophy and dysfunction later in life in India. There is evidence that low birth weight followed by weight gain in adolescence is associated with several cardiovascular disease risk factors in adulthood. However, there is less evidence of associations with cardiac structure and function, in particular in South Asians, which serves as the motivation for the planned study. The study is based on two existing Indian birth cohorts, the The New Delhi Birth Cohort and the The Vellore Birth Cohort. As part of the planned study, a further round of measurements will be collected on expected 3,000 members of the cohorts with earlier follow-up measurements available. The cohort members are now between 43-50 years old. The data to be collected include the cardiac structure outcome measurements using echocardiography. Power calculations for the planned association analyses are presented. Linear and logistic regression and conditional growth models are planned for analysis. Substantial attrition of the original birth cohorts, mainly due to death and migration, is mentioned as a limitation of the study. However, it is noted that the the remaining cohort members are similar to the original cohorts in terms of their early life characteristics. The study is funded and has ethics approval. I have just one question: is there evidence for the suggested participation rate (3,000 of the 3,500 re-contacted)?

Authors' response: The suggested participation rate is based on an initial tracing exercise in early 2016, before initiating IndEcho. Both the centres together were able to contact 3,500 individuals out of the total 3,664 alive participants from the adult Phase 1 follow-up (NDBC: 1,508 and VBC: 2156). From this figure, and based on our past experience of follow-up in these cohorts, we expect approximately 1,250 from NDBC and 1,750 from VBC (total 3000) to participate in the IndEcho study.

VERSION 2 – REVIEW

REVIEWER	David Burgner Murdoch Children's Research Institute Melbourne Australia
REVIEW RETURNED	20-Jan-2018

GENERAL COMMENTS	The authors have undertaken a detailed and thoughtful revision of the manuscript in response to the detailed reviewers' comments. The manuscript is much clearer as a result and I have no further major comments. A summary sentence at the end of the abstract outlining the potential outcomes and significance might be helpful. I think that the term carotid intimal-media thickness (rather than intimal) is more usual. I look forward to the findings from this important study.
--

REVIEWER	Jingyan Tian Ruijin Hospital affiliated to Shanghai Jiaotong University School of Medicine, Shanghai, China
REVIEW RETURNED	03-Feb-2018

GENERAL COMMENTS	I have no further comment.
----------------------------

REVIEWER	Olli Saarela University of Toronto, Canada
REVIEW RETURNED	06-Feb-2018

GENERAL COMMENTS	No further comments.
----------------------